

# The R package 'eseis' – a comprehensive software toolbox for environmental seismology

Michael Dietze[1]

[1]GFZ German Research Centre for Geosciences, Section 5.1 Geomorphology, Potsdam, Germany

*Correspondence to:* Michael Dietze (mdietze@gfz-potsdam.de)

**Abstract.** Environmental seismology is the study of the seismic signals emitted by Earth surface processes. This emerging research field is at the seams of seismology, geomorphology, hydrology, meteorology, and further Earth science disciplines. It amalgamates a wide variety of methods from across these disciplines and, ultimately, fuses them in a common analysis environment. This overarching scope of environmental seismology asks for a coherent, yet integrative software, which is accepted by many of the involved scientific disciplines. The statistic software R has gained paramount importance in the majority of data science research fields. R has well justified advances over other, mostly commercial software, which makes it the ideal language to base a comprehensive analysis toolbox on. The article introduces the avenues and needs of environmental seismology, and how these are met by the R package 'eseis'. The conceptual structure, example data sets and available functions are demonstrated. Worked examples illustrate possible applications of the package and in depth descriptions of the flexible use of the functions. The package is available under the GPL license on the Comprehensive R Archive Network (CRAN) and maintained on Github.

## 1 Introduction

Environmental seismology exploits the seismic signals emitted by Earth surface processes, the processes that shape our planet (Larose et al., 2015; Burtin et al., 2016). This new research field emerged as more and more sensitive, mobile and flexible sensors and data loggers became available. The field was pioneered by studies of mass wasting and fluvial process in high mountain landscapes, explicitly bridging seismological expertise with key geomorphic interests (e.g. Helmstetter and Garambois, 2010; Burtin et al., 2010; Dammeier et al., 2011; Hibert et al., 2011). In parallel, the foundations were laid for physical models relating the measured seismic signals to underlying physical processes (e.g. Tsai et al., 2012; Gimbert et al., 2014; Farin et al., 2015). This increasingly allows applications of the technique for early warning and rapid response aspects (Mainsant et al., 2012; Hibert et al., 2014; Zeckra et al., 2015), for investigating event trigger processes (Dietze et al., 2017a), for monitoring extremely active landscapes (Bartholomaus et al., 2012; Burtin et al., 2013; Schöpa et al., 2017), but also for exploring the precision and limitations of the seismic method (Dietze et al., 2017b).

Seismic data processing tools have reached an advanced and diversified level. There is a wealth of encapsulated programs, software libraries and script language routines available (e.g. IRIS, 2017a; ETH, 2017, and links therein). Due to their structure, compiled software solutions can usually only provide very limited interaction capabilities with other, generic data analysis tools





that can be utilised for exploratory, explanatory and inverse data manipulation beyond the designed work focus of the initial software. Furthermore, since environmental seismology integrates several neighbouring and more distant scientific fields, to which the seismological results are passed as input data, it is essential to find a common language. This common language constraint applies to both, the scientific jargon and the data analysis software language. Ideally, this common software language

should be open, transparent and user-driven (cf. chapter 2), flexible and extendible, it should have a wide acceptance in different fields of science. It should be easy to learn and support efficient and fast computation of large and diverse data sets. There is one software language tailored to data science that fulfils all these qualifiers: the free statistic software R (RCoreTeam, 2015). Across scientific disciplines and on the commercial market, R has gained world wide acceptance as robust, innovative and flexible software and language (Tippmann, 2014). This is reflected by a rich number of online tutorials, text books, webinars,

the utilisation of R in academic teaching courses, the predominant role R plays in medical and life sciences, and the more than 10000 packages hosted on the Comprehensive R Archive Network (CRAN).

Besides these software-structural points, there are further arguments for providing new functionalities for the wider environmental seismology purpose rather than working with existing tools. Software, tailored to signals emitted by Earth surface processes require other numeric approaches to problems like seismic event picking and classification or seismic source lo-

cation than are expected in a classic seismology scope. These other approaches need to be provided without compromising the existing functionalities. Additionally, the environmental seismology community produced a series of innovative models to relate Earth surface dynamics to the seismic signals they generate (e.g. Tsai et al., 2012; Gimbert et al., 2014) or to changes in the medium in which the seismic waves travel (Sens-Schönfelder and Larose, 2010; Larose et al., 2015). These models and approaches should be fused into a coherent software environment rather than staying published as isolated programs or pseudo

code definitions to allow their seamless inclusion in interdisciplinary studies.

## 2  Environmental seismology and open science

Reproducibility of scientific research is a key goal, identified across disciplines (e.g. Lane, 2014; David et al., 2016; Munafó et al., 2017). The foundations of reproducible research are complete access to both, the data and software used to generate the results. The former is increasingly demanded by research funding agencies and journals, usually in the form of data repositories

including allowance of access embargo periods of some months. This data policy is especially relevant for seismic data usable for investigating environmental dynamics, because the data can be used in many different purposes, beyond the initial research scope. However, the other requirement, open software, is less often provided. Proprietary software with license costs may be a minor issue within universities of western societies but becomes a major obstacle for less developed countries.

Another aspect of reproducibility with respect to software concerns the life time, computer platform support and version of

the analysis software (Kreutzer et al., 2017). Ideally, the calculations should be possible and yield the same results across all major computer platforms (Windows, Linux, Mac OS) and across different versions of the software. Since the latter requirement is difficult to maintain and may be in conflict with the actual purpose of software updates, there should at least be some information on the version with which the results were obtained and the possibility to use exactly this software version for



reproducing previous work. R with its package policy (RCoreTeam, 2015) is a pillar of stability in this respect; all versions of a package are kept in online accessible archives and community-driven projects such as 'docker' (Karambelkar, 2017) and 'packrat' (Ushey et al., 2016) allow version-specific processing environments.

Processing seismic data involves a long work flow with many processing steps, each requiring individual parameter adjust-

ments. Thus, not only data availability and open source software are key requirements for reproducible science, but also a full documentation of the processing chain. To secure reproducibility and transparency the software should contain a processing documentation scheme. There are ways to combine continuous explanatory text with code snippets and the results of its evaluation (e.g., R notebooks, R markdown files, Jupyter notebooks), which can be provided as supplementary materials to publications. However, a more robust way would be an implicit one that does not require the scientist to take care of migrating

the analysis steps into a full documentation, manually.

Documentations of functions should be coherent and systematic. While scripts that circulate among working groups may be adequately documented in some cases, it is a rather common phenomenon that updates or changes in the code diffuse into different script versions and branches, if not coordinated centrally. Likewise, the practice of commenting and documenting is far from being a standard. To avoid such pitfalls, R and its package strategy have rigorous standards that no package can cir-

cumvent (RCoreTeam, 2015). A package must always contain a concise and detailed description document, plain text function definitions and optional additional low level code, a reference documentation, example data sets and tested examples for each function. A package undergoes extensive automatic and manual tests before it is accepted by the CRAN and packages like 'roxygen2' (Wickham and Chang, 2017) allow automated builds or updates of the documentation from within the function source code. In a similar manner, incremental maintaining and updating a package needs to follow the same systematic and rigorous

criteria as a submission to the CRAN. Git and Github provide all the essential functionalities to pursue this goal (Chacon and Straub, 2013).

## 3 The package contents

The R package 'eseis' version 0.4.0 will be made available on the CRAN with the publication of this manuscript. Development is handled via Git and the respective latest developer version is hosted on Github. Information about updates, content,

applications and bugs is hosted on a package website (http://micha-dietze.de/pages/eseis.html). The package contains 51 functions and two example data sets. It makes use of a series of dependency packages, namely 'caTools' (Tuszynski, 2014), 'fftw' (Mersmann, 2017), 'IRISSeismic' (Callahan et al., 2017), 'matrixStats' (Bengtsson, 2017), 'methods' (R Core Team, 2017a), 'multitaper' (Rahim et al., 2014), 'raster' (Hijmans, 2017), 'Rcpp' (Eddelbuettel et al., 2017), 'rgdal' (Bivand et al., 2017), 'signal' (Ligges et al., 2015), 'sp' (Pebesma and Bivand, 2017) and 'XML' (Temple Lang and CRAN-Team, 2017).

Installing the package from CRAN is perhaps the most convenient way (`install.packages("eseis")`). However, to get the latest developer version and other branches, the Github repositories need to be used. With the package 'devtools' (Wickham and Chang, 2017) is is easy to do this in R by running one line of code:

```
devtools::install_github(repo = "coffeemuggler/eseis", ref = "0.4.0")
```



Some platforms require prior installation of R-independent software such as Java runtime environment, openGL, fftw or gdal. Once all these independent dependencies and the required packages as listed above are installed, 'eseis' can be installed.

The package allows all typical base functionalities for a seismic data analysis work flow: data import/export, signal processing/preparation, event picking, seismic source location, spectrum and spectrogram computation, and plot generation (fig 1).

Further functions allow seismic data download from available data bases and conversion of raw measurement files to a coherent file structure, picked up by other functions. Thus, 'eseis' is clearly distinct from other available R packages that mainly focus on data import and quality control ('IRISSeismic' (Callahan et al., 2017)) or earthquake analysis ('RSEIS' (Lees, 2017) and its interlinked packages).

## 3.1   General package philosophy

**The eseis object paradigm**. In the 'eseis' package seismic data is handled as a S3 object of class `eseis` (e.g., `x`), which are basically a list objects with four elements: the signal vector (`x$signal`), a list with meta information (`x$meta`), another list with the raw header information of the imported file (`x$header`) and a list containing the processing history the `eseis` object has been subject to (`x$history`, see below). Thus, whatever the file format of the imported data may be (cf. section 3.2.1), it is always represented in the same way within the 'eseis' environment. Although it would be possible from a computational

point of view to handle only the signal vector of an imported seismic file, it is much more fail-safe, coherent and convenient to work with the `eseis` object. In many processing steps it is necessary to access or even modify object attributes such as the start time (`x$meta$starttime`) and number of samples (`x$meta$n`) or the sampling interval (`x$meta$dt`). The functions of the package automatically query or update these attributes if needed so that changes in any of the parameters do not need to be considered.

**The processing history** to which an imported seismic data set has been subject is recorded in the `eseis` object itself (`x$history`). This allows full access to the work flow that lead to the values the object contains at a given time. Upon creation of an `eseis` object, due to importing a seismic file, two base list elements are created. The first one documents the computer system configuration under which the object has been created (`x$history[[1]]`). It includes among others the R version, the computer platform and operation system, local settings such as UTF encoding and time zone, and the other

attached R packages in their current version. The second list element (`x$history[[2]]`) documents the actual file import step. Like for any other subsequent processing function the documentation includes the time stamp of the function call, the function call itself, all function arguments with the respective values, and the duration of the function operation. With each subsequent processing step a new history list element is appended. This feature provides all information to reproduce all results. The function `write_report` can be used to generate html-based report documents of `eseis` objects, which can be stored

for traceability purpose, shared with colleagues or published along with the raw data in articles.

**The function naming scheme** follows a higher-level convention. There are currently eight groups to which all functions belong and that appear in the function name: read and write seismic files (`read_` and `write_`), plot data and data derivatives (`plot_`), signal processing (`signal_`), spatial data handling and source location (`spatial_`), lookup tables and parameter lists (`list_`), model definitions (`model_`) and auxiliary functions for convenient data management (`aux_`). This naming





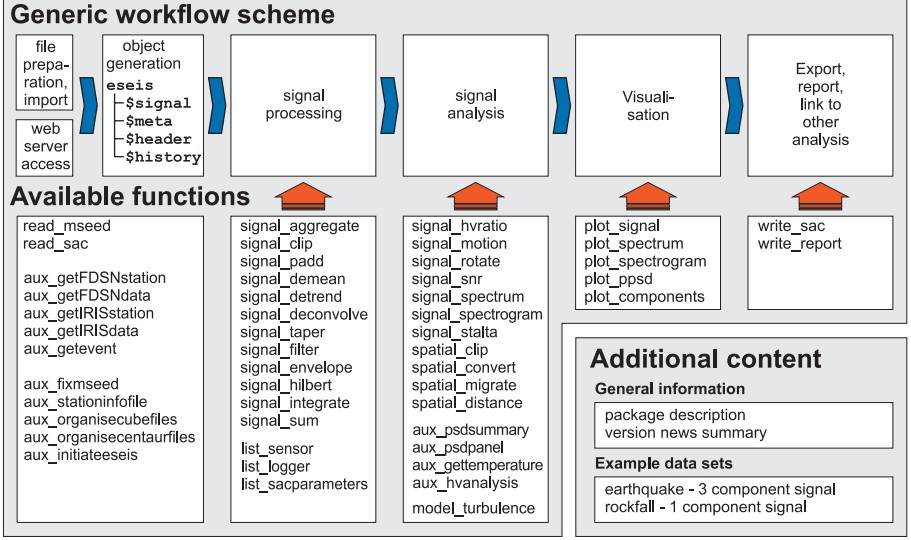

**Figure 1.** General work flow and available functions of the 'eseis' package.

scheme is partly due to the R package and function documentation convention, which basically is a reference documentation where a manual is compiled by sorting all functions of a package in alphabetic instead of semantic order. The naming scheme is also designed to minimise so called masking effects, multiply assigned function names masking each other when several packages are loaded. Although it is in principle easy to point at the right function by adding the package name and two colons

5   in front of the function name (e.g., `eseis::read_sac`) this is not intuitive and would be in conflict with several of the premises defined in the introduction and section 2.

**Full support of and access to native R functionalities** is a further key objective of the 'eseis' package. This means that `eseis` objects can be passed to all efficient data manipulation approaches in R such as piping with the 'magrittr' package (Bache and Wickham, 2014), vectorised manipulation of lists of `eseis` objects with `lapply` and multi-core data manipula-

10  tion with `parLapply` on computers with more than one CPU. Access to R functionalities implies that almost all functions are written in R, without low level programming languages. This allows for full insight to the function source code, a pillar of the transparency constraint and the precondition for any user-based modification of functions. The only exception from this rule is the function `signal_stalta`, which makes use of C++ to significantly improve computational performance in the necessary `for` loop routine, yielding a speed increase of about factor 200. In all other cases, function source code was written

15  in vectorised form to optimise performance.

## 3.2   Available functions

The 'eseis' package is designed for full processing routines of seismic data, from the import of raw files to data preparation, analysis, presentation/visualisation and export or passing to further R and external functionalities. Additionally, more com-



plex, predefined work flows are implemented, such as converting non standard seismic file formats to mseed or sac files and organising them in a coherent file structure, preparing overview spectrogram panel plots for a seismic network throughout the station deployment time period, or importing all seismic traces of a given event based on time, channel (i.e., an orthogonal spatial signal component), and stations to include. A documentation of all functions and example data sets as mandatory part

of the package compilation process is available as PDF document on the CRAN. However, 'eseis' is not designed as substitute for classic seismic data analysis tools such as SAC or Obspy (Beyreuther et al., 2010), which may be much more appropriate when the main tasks include having full support of all relevant seismic data file formats, full access to XML-SEED, and classic seismology research fields.

### 3.2.1 Data import

Supported seismic file types are SAC (IRIS, 2017b) and miniseed (IRIS, 2012), implemented by the functions `read_sac` and `read_mseed` (fig. 1). By default, the import will yield an eseis object, unless the user wishes to work with a separate signal and time vector (`eseis = FALSE`). Thus, regardless of the input file format, the resulting object is a homogeneous representation of the signal and its meta data. It is always possible to specify several seismic files in consecutive order to obtain a continuous trace in R (`append = TRUE`), otherwise, each file will be imported separately to yield a list of the individual

traces). It is possible to select the individual object elements to be imported or omitted (e.g., `signal = FALSE`), which is useful to just screen the header or meta information of seismic files instead of reading the entire signal vector. The function `read_mseed` wraps functions of the package 'IRISSeismic' (Callahan et al., 2017) to prevent redundant coding efforts.

A more convenient and robust way to import seismic data from local files is provided by the function `aux_getevent`. It requires specifying a time period, the channel(s) and one or more station IDs, and reads all data into a coherent list ob-

ject, which can be directly used for further signal processing steps. However, the function requires the data to be stored in a predefined directory structure (fig. 2). The files must be stored in directories organised by year and Julian day. In each Julian day directory, an hourly seismic file must be named with station ID, Julian day, hour, minute and second of the start time of the seismic signal, seismic channel name, and file extension (fig. 2). This structure and naming convention is the base for `aux_getevent` and many other functions of the `aux_` family. If future avenues of research suggest further structure and

naming schemes, these can be implemented, as well. The easiest way to bring files into the described structure is to use the functions `aux_organisecentaurfiles` and `aux_organisecubefiles`. The former is originally written for seamless conversion of files recorded by a Nanometrics Centaur data logger but can also be used with other loggers if these store the data in an appropriate format and structure (see function documentation for details). The latter function is designed for files recorded by Omnirecs DataCube loggers and handles the complete work flow of converting daily cube files to mseed files,

cutting them to hourly segments, optionally further to SAC files, and establishing the directory structure as described above.

Apart from local seismic files it is possible to work with seismic data from web-based services, such as IRIS or FDSN dataselect. These options are implemented by `aux_getIRISstation`, `aux_getIRISdata`, `aux_getFDSNstation` and `aux_getFDSNdata` (fig. 1). The basic principle behind both, the IRIS-based and the FDSN-based functions is that first one needs to identify seismic stations that are present within a given radius around a location defined by latitude and longitude.





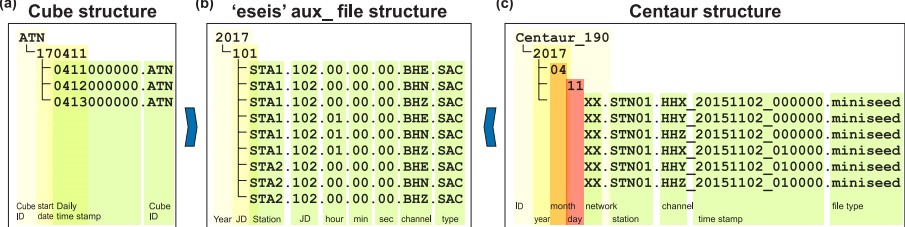

**Figure 2.** File organisation from commonly used data loggers, Omnirecs DataCube (a) and Nanometrics Centaur (b), and how they are converted to the standard format used by most of the auxiliary functions of the 'eseis' package. After conversion, seismic files are organised after year and Julian day directories and, in these, as file names denoting station ID, Julian day, hour, minute, second, channel and file name extension.

With the station data (the sncl code for `aux_getIRISdata` and network ID and station code for `aux_getFDSNdata`) at hand, the mseed data sets can be temporarily downloaded and imported as `eseis` object.

### 3.2.2 Data processing

The next steps after the import of seismic data typically include signal processing in order to prepare the data for the actual
analysis part. In most cases it is necessary to correct the signal for the instrument response, i.e. signal deconvolution, performed by the function `signal_deconvolve`. The main arguments needed for the deconvolution are the characteristics of the sensor (poles, zeros, generator constant or sensitivity and the normalisation factor) and logger (AD conversion factor). Additional arguments can be set if appropriate (`gain` correction, `p` signal taper proportion, `waterlevel` value to avoid division by zero). The deconvolution step differs from the approach implemented in, for example, Obspy (Beyreuther et al., 2010), where
dataless seed files are the primary carrier of parameter information.

The deconvolved signal can be subject to manipulations like removing the mean (`signal_demean`) or linear trend (`signal_detrend`), tapering by proportions of the signal length or numbers of samples (`signal_taper`) to remove edge effects, aggregating or decimating (`signal_aggregate`) to decrease the sampling density, padding with zeros to the next higher order of $2^n$ (`signal_padd`) for more robust and efficient Fourier transform operations, integrating (`signal_integrate`)
to calculate displacement from velocity, calculating the vector sum of signal components (`signal_sum`), calculate the Hilbert transform (`signal_hilbert`) and the signal envelope (`signal_envelope`), and filtering the signal in the time domain (`signal_filter`). It is also possible to clip an imported signal to a given time interval (`signal_clip`).

With an appropriately prepared (`eseis`) object signal analysis can be performed. Available functionalities allow picking events based on the classic STA/LTA algorithm (Allen (1982), `signal_stalta`), which is implemented as C++ code, calcu-
lating spectra (`signal_spectrum`) and spectrograms (`signal_spectrogram`), the signal-to-noise ratio (`signal_snr`), rotate the signal components (`signal_rotate`), infer particle motion analysis (`signal_motion`) and calculate the horizontal to vertical component ratio (`signal_hvratio`). Special emphasis is on spatial analysis to locate environmental



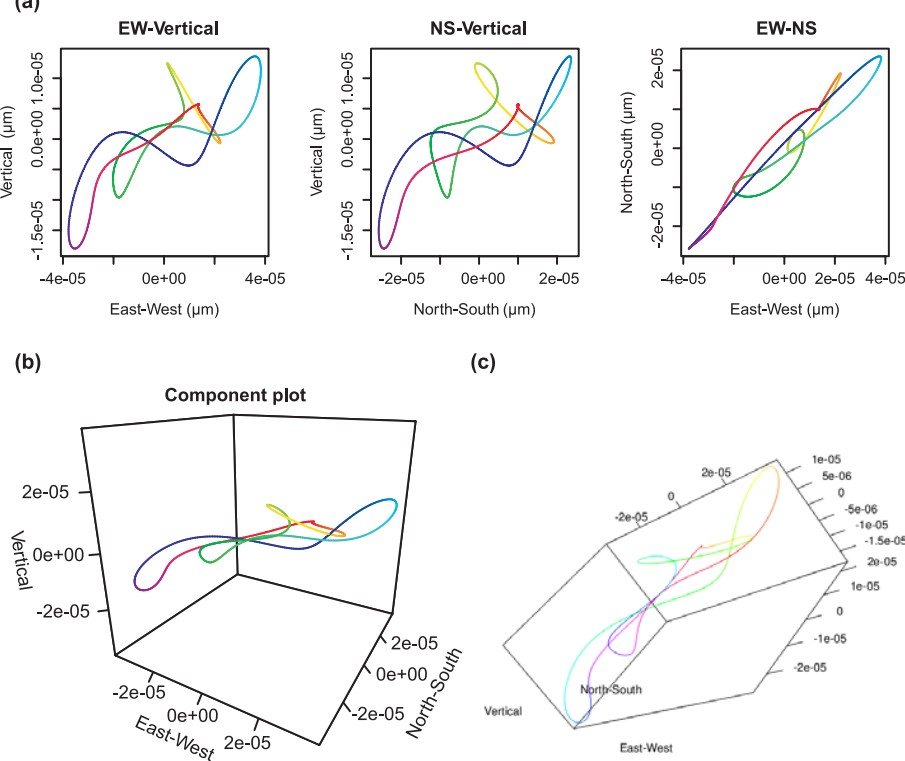

**Figure 3.** Plot types to visualise particle motion. (a): 2D plots of the components, (b): 3D perspective plot, (c): interactive scene that can be rotated and zoomed.

signals, one of the main application fields of environmental seismology (`spatial_migrate`, (cf. Burtin et al., 2016; Dietze et al., 2017a), along with helper functions to prepare spatial data (`spatial_clip`, `spatial_convert`, `spatial_distance`). For more complex but routine analysis tasks, there is a growing suite of auxiliary functions. These include the possibility to generate overview panels of spectrograms, either for constant time intervals across all stations in a seismic network

5  (`aux_psdpanel`) or for different time aggregation levels but a single seismic station (`aux_psdsummary`). The function `aux_hvanalysis` performs a comprehensive horizontal to vertical signal ratio analysis for given time slices and generates a graphical output of the results.

Visualisation of the data is key for the final presentation of the results. R offers powerful methods for high quality plot output, though the scripting effort for user-adopted plots can be fairly high (e.g. Dietze et al., 2016) and the learning curve is steep.

10  Thus, all plot functions in the 'eseis' package are set up with meaningful default arguments to minimise the amount of overhead code to generate a fairly pretty plot. Likewise, the massive amount of data to handle (more than 17 million samples per day for a single component recorded at 200 Hz) can easily bring the default plot functions of R to a limit. Thus, the plot functions in the 'eseis' package were optimised for speed and aggregation of data beyond visible effects. The package allows plotting the

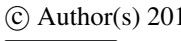



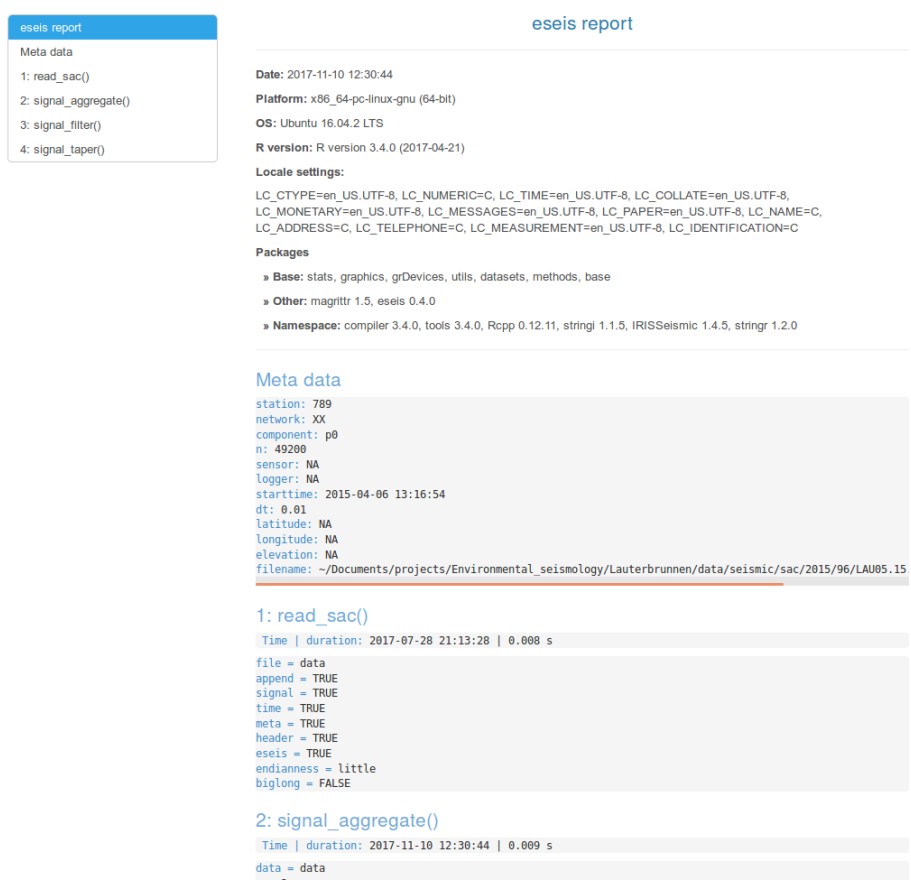

**Figure 4.** Section of the html-based report output. The interactive document lists the computer and software environment under which the processing took place, as well as all manipulation steps the data set underwent, including function calls and argument values.

signal waveforms (`plot_signal`), their spectra (`plot_spectrum`), calculated spectrograms (`plot_spectrogram`) and probabilistic spectra (`plot_ppsd`), and the three signal components as 2D-plot, 3D-plot and interactive rotatable 3D scene (`plot_components`, cf. fig. 3).

The export of seismic data is not one of the key features of the 'eseis' package. Currently, writing functionality is only implemented for the SAC file format (`write_sac`), which however, can be easily converted to different formats by other software (e.g. Beyreuther et al., 2010). To enable full reproducibility of any analysis step in the package (cf. section 3.1), the function `write_report` generates a summarising html-based report of the manipulation history of the `eseis` object (cf. fig. 4).

In order to use the numeric data for further analysis in R or to generate used-adopted plots, every function (except for some plot functions and the report function) returns standard R objects, either as lists of class `eseis` or numeric matrices, vectors, and spatial data objects.



## 3.3 Example data sets

The 'eseis' package contains two example data sets of minimum size that are used to illustrate the functionalities of the package. One data set contains the seismic signal of a rockfall preceded by a small earthquake recorded by the vertical signal component. The other data set contains a typical earthquake recorded by thee spatial components. Both data sets were recorded by a temporary network in the Lauterbrunnen Valley, dedicated to analysing rockfall activity and its trigger conditions (i.e. event ID 30 in Dietze et al., 2017a). The network consisted of Nanometrics Trillium Compact TC120s sensors, being sampled by Omnirecs DataCube[3]ext loggers at 200 Hz with a gain of 1.

## 4 Worked examples

Two worked examples are discussed to illustrate the capabilities and typical work flows of the 'eseis' package. With the focus on package utilisation, study site description and environmental interpretation of the results are kept minimal. Both worked examples are provided as R markdown documents in the supplementary materials.

A typical session or script initiation contains loading the package, setting the workspace (the directory in which all data will be searched and stored by default), and reading the station info table, i.e. a table that contains essential information about the seismic stations (cf. table 1).

```
## load package
library("eseis")

## set working directory
setwd(dir = "~/data/seismic/")

## load station info table
stations <- read.table(file = "stations.txt",
                       header = TRUE,
                       stringsAsFactors = TRUE)
```

## 4.1 Cliff coast collapses

During a pilot study, the cliff coast section of Germany's largest island, Rügen, has been instrumented by four Nanometrics Trillium Compact 120s broadband seismometers and Omnirecs Cube[3]ext data loggers, recording at 200 Hz and powered by 70 Ah batteries. The stations were deployed between April and May 2017 on top of the about 100 m high chalk cliffs, spaced by about one km, in hand-dug pits of about 50 cm depth. The primary purpose for this monitoring campaign was to detect and locate cliff collapses that occur as rock falls, rock avalanches, rotational slides and debris flows (LUNG, 2003).





**Table 1.** Station information table, containing all relevant meta data for the deployed seismic stations. Coordinates x and y are given in UTM coordinates (zone 33N), station z and station d denote station elevation (m asl.) and sensor deployment depth (m).

| ID | name | x | y | station z | station d | sensor type | logger type | sensor ID | logger ID |
|----|------|---|---|-----------|-----------|-------------|-------------|-----------|-----------|
| RUEG1 | Open Backyard | 413107.2 | 6048242.7 | NA | 0.5 | TC120s | Cube3ext | NA | ARV |
| RUEG2 | Beloved Peregrine | 414441.7 | 6046841.3 | NA | 0.5 | TC120s | Cube3ext | 2796 | ART |
| RUEG3 | Shrapnel City | 414189.2 | 6045155.2 | NA | 0.5 | TC120s | Cube3ext | 2763 | ARS |
| RUEG4 | Running Fishermen | 413969.6 | 6043145.7 | NA | 0.5 | TC120s | Cube3ext | 2797 | ARU |

### 4.1.1 Raw data handling

Data processing starts with copying the raw files from the data loggers to a local directory in a way depicted in fig. 2 a and creating the station info file (cf. table 1). Specifically, it is a file denoting station ID, full station name, longitude (easting), latitude (northing), elevation, sensor installation depth, sensor type, logger type, sensor ID and logger ID. These information can either be collated manually or by using the function `aux_stationinfofile`, which extracts most of the relevant data from the cube files. The function can be run on a given fraction of CPUs (e.g., `cpu = 0.5`) and the number of cube files to be analysed for extract the station coordinates can be set, as well (e.g., `n = 11`). The function creates the station info ASCII table and, if enabled by the user, files with the GPS data.

```
aux_stationinfofile(name = "station_info_RUEG17_network_dd",
                    input_dir = "../cube/",
                    output_dir = "../sac/",
                    station_ID = c("RUEG1", "RUEG2",
                                   "RUEG3", "RUEG4"),
                    station_name = c("Open Backyard", "Beloved Peregrine",
                                     "Shrapnel City", "Running Fishermen"),
                    station_d = rep(x = 0.5,
                                    times = 4),
                    sensor_type = rep(x = "TC120s", times = 4),
                    logger_type = rep(x = "Cube3ext", times = 4),
                    sensor_ID = c(NA, 2796, 2763, 2797),
                    logger_ID = c("ANT", "ANN", "ALF", "ANV"),
                    unit = "dd",
                    gipptools = "~/software/gipptools-2015.225")
```





Based on the station info file the data can be transformed from the initial structure to the one shown in fig. 2 b. This step is done by the function `aux_organisecubefiles`. In order to use the function the GIPPtools software (Lendl, 2017) has to be installed.

```
aux_organisecubefiles(stationfile = "station_info_RUEG17_network_dd.txt",
5                      input_dir = "../cube/",
                       output_dir = "../sac/",
                       format = "sac",
                       gipptools = "~/software/gipptools-2017.013/",
                       verbose = TRUE)
```

The user can decide if the output files shall be mseed or sac files (e.g., `format = "sac"`), which naming convention shall be used (e.g., `channel_name = "p"`), and how many CPUs shall be used. The GIPPtools offer further options to control how to handle samples outside of GPS time stamps (`fringe`) and the degree of command line processing information (`verbose`). In more recent Java Runtime Environment versions, on which the GIPPtools are based, there has been a problem with the heap space, which lead to crashes of the GIPPtools during Cube file conversion. This shortcoming can be accounted
for by increasing the value of `heapspace`. Likewise, the function accepts manually converted mseed files to be organised in the output structure (`mseed_manual = TRUE`). Otherwise, the function converts all Cube files to daily mseed files, clips these to hourly files, imports them to R, appends the meta data, creates the output directory structure and saves the files in that structure in the desired file format.

### 4.1.2   Event data import

Seismic data of events can be imported either by providing the relevant seismic file names, importing them and clipping them to the relevant time window (e.g., 10 sec), which can be done manually, in a loop (not advised and not shown) or as a list approach:

```
## define files to import
files <- c("2017/080/RUEG1.17.80.04.00.00.BHZ.SAC",
25          "2017/080/RUEG2.17.80.04.00.00.BHZ.SAC",
            "2017/080/RUEG3.17.80.04.00.00.BHZ.SAC",
            "2017/080/RUEG4.17.80.04.00.00.BHZ.SAC")

## manual import
x <- list(RUEG1 = read_sac(file = files[1], eseis = TRUE),
              RUEG2 = read_sac(file = files[2], eseis = TRUE),
              RUEG3 = read_sac(file = files[3], eseis = TRUE),
```



```
            RUEG4 = read_sac(file = files[4], eseis = TRUE))

  ## list-based import and clipping
  x <- lapply(X = files,
5             FUN = read_sac,
              eseis = TRUE)

  names(x) <- stations$ID

10 ## clip signal to event
  x <- signal_clip(data = x,
                   limits = as.POSIXct(x = c("2017-03-21 04:38:50",
                                             "2017-03-21 04:39:00"),
                                       tz = "UTC"))
```

15    Another way, `aux_getevent`, would combine the `read_sac` and `signal_clip` calls from above but also account for problems such as time limits out of the data range or events covering more than a full hourly seismic file. The event data can be imported by the following code, where `station$ID` denotes the seismic station IDs of the station info table (table 1):

```
## load event data
x <- aux_getevent(start = as.POSIXct(x = "2017-03-21 04:38:50",
20                                     tz = "UTC"),
                  duration = 10,
                  station = stations$ID,
                  component = "BHZ",
                  dir = "sac/",
25                eseis = TRUE)

## simplify structure due to presence of only one component
x <- lapply(X = x, FUN = function(x) {x[[1]]})
```

Either way, the result is a list containing the vertical components (`"BHZ"`) of all four seismic stations for the given time in-
30 terval. For further signal inspection, analysis and interpretation one usually corrects for the instrument response. The respective function `signal_deconvolve` needs additional information about characteristics of the seismic sensor and data logger (cf. 3.2.2), the logger gain, the fraction of the signal that will be tapered during the deconvolution process (by default $10^{-6}$) and a water level value (by default $10^{-6}$):

```
## deconvolve signals
```



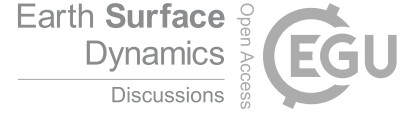

```
x <- signal_deconvolve(data = x,
                       sensor = "TC120s",
                       logger = "Cube3extBOB")
```

In case that a sensor or data logger is not contained in the list, provided with the package, the example part of the function

5  documentation shows how to use user defined deconvolution parameters.

### 4.1.3  Data processing

The vertical component signals of the four stations can now be detrended, filtered, tapered and, for example plotted (fig. 5 a).

```
## detrend signals
x <- signal_detrend(data = x)

## filter signals
x <- signal_filter(data = x,
                   f = c(5, 10))

## taper signal based on n samples
x <- signal_taper(data = x,
                  n = 300)

## plot signal waveforms
lapply(X = x, FUN = plot_signal)
```

### 4.1.4  Data analysis

Likewise, one can calculate spectra and spectrograms (time evolution of spectra calculated within time windows) from the signals. Any spectrum is calculated using functionalities from the package 'stats' (R Core Team, 2017b), namely `spec.pgram` (spectral density is estimated based on a smoothed periodogram) and `spec.ar` (spectral density is estimated from an autore-

25  gressive fit). Both functions have further arguments, for example to pad the time series with zeros and remove the mean, which can be passed through `signal_spectrum` using the `...` argument, if appropriate. The default option of `signal_spectrum` is the periodogram option, whereas the autoregressive variant results in a generally smoother spectrum (fig. 6). Additionally, for very short signals, it is possible to utilise the multi taper option (`multitaper = TRUE`, Thomson (1982)). However, this increases computation time significantly. As bridge between time domain (signal waveform) and frequency domain (signal

30  spectrum), there are spectrograms. There, a signal is cut into optionally overlapping windows and spectra are calculated for each of these windows. Thus, spectrograms represent the spectral evolution of a signal with time. Welch (1967) introduced the



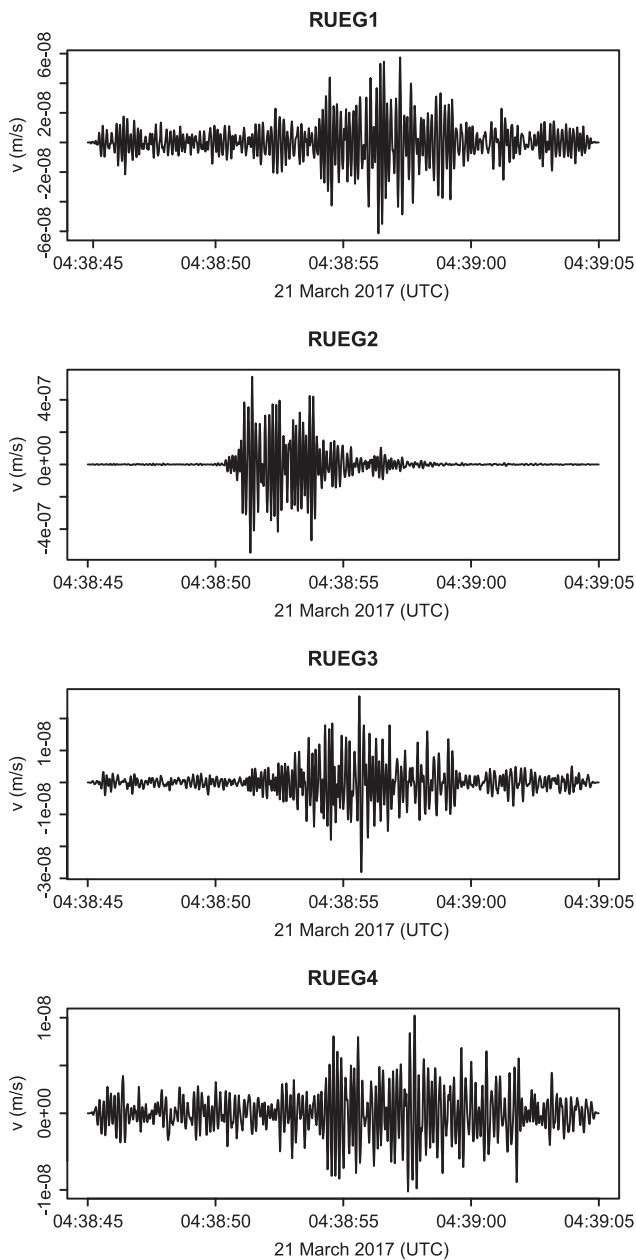

**Figure 5.** Seismic waveforms of a cliff coast collapse as recorded by four stations in the Jasmund National Park. Signals have been deconvolved, detrended, filtered between 5–8 Hz and tapered by 300 samples. Note both, the time offset and amplitude decrease as the event is recorded further away from its origin about 80 m east of station RUEG2.

idea to calculate spectra for each window based on sub-windows, that again may or may not overlap within a window. Thereby,





the spectra are averaged and result in a more robust and smoothed representation. This option is implemented by the option `Welch = TRUE`. Likewise, multi tapers can be utilised for spectrograms (`multitaper = TRUE`), again keeping in mind that this results in significant additional computation time.

```
## calculate a spectrum using the autoregressive option
s_2 <- signal_spectrum(data = x$RUEG2,
                       method = "autoregressive")

## calculate a spectrogram using the Welch option
s_3 <- signal_spectrogram(data = x,
                          Welch = TRUE,
                          window = 1.0,
                          window_sub = 0.7,
                          overlap = 0.95,
                          overlap_sub = 0.95)
```

### 4.1.5   Plotting

Plotting spectral data sets is possible by designated functions. `plot_spectrum` is used for spectra. Note how the periodogram-based spectrum (fig. 6 a) creates a much rougher graph than the autoregressive option (fig. 6 b). The spectrogram (`plot_spectrogram`, fig. 6 c) shows the onset of the cliff collapse and how it is dominated by frequencies below 20 Hz that give way to prolonged activity across the full recorded frequency spectrum after about 5 seconds. The probabilistic spectrogram (`plot_ppsd`, fig. 6 d) depicts how the collapse event shifts the entire frequency spectrum by about 30 dB towards higher power, especially in the 5 to 15 Hz range.

### 4.1.6   Source location

Locating the source of a seismic signal is one of the most relevant goals in environmental seismology. However, classic seismological approaches fail or at least need modification to be appropriately applied to the seismic signals emitted by Earth surface processes (e.g. Lacroix and Helmstetter, 2011; Burtin et al., 2013; Dietze et al., 2017a). Currently, the 'eseis' package contains a full waveform or signal envelope migration approach that performs a grid-search to find the location in space that best explains the overall time offsets between event signals as recorded at pairs of stations. The algorithm is based on methods described by Burtin et al. (2013, and references therein). Location of a seismic source with this approach requires a digital elevation model (DEM) depicting the topography of the area within which the source shall be searched. The DEM is used to generate distance maps, i.e., lookup tables of topography-corrected surface distances between a seismic station and any pixel. These distances are later converted to travel times using average apparent surface wave velocities and the pixel with the best overall travel time value explaining the time offsets between stations pairs is deemed to be the most likely source location. The





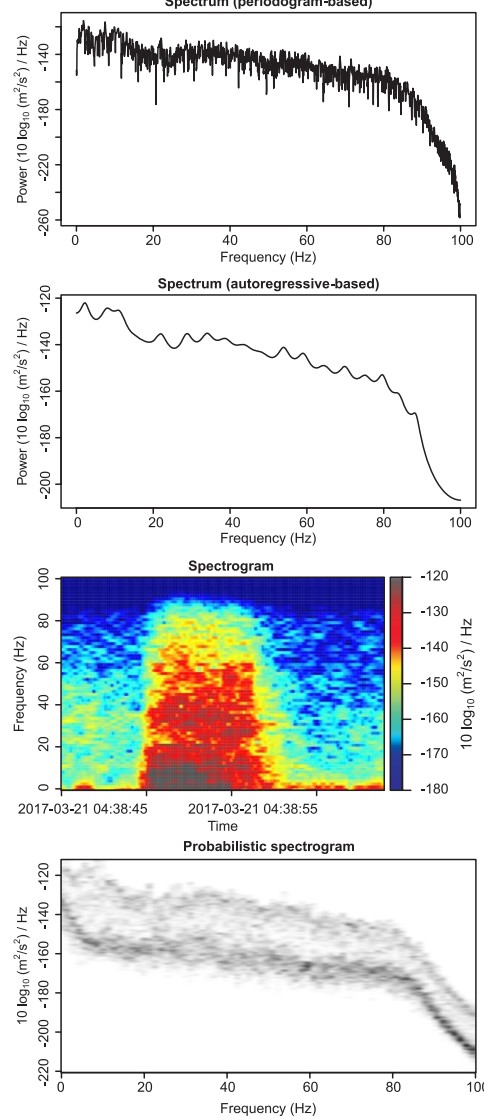

**Figure 6.** Spectra (a and b), spectrogram (c) and probabilistic spectrogram (d) of the cliff collapse event depicted in figure 5, station RUEG2.

distance maps are created with the function `spatial_distance`, which requires the DEM data set and the seismic station coordinates, usually delivered with the station info file, and returns the distance data sets and the inter-station distances as a list object for later use. The calculation of the lookup tables is computationally extensive and can take hours for grid with some ten thousand pixels. It is thus useful to save the output of the function for efficient later use.

```
## load DEM using the 'raster' package
dem <- raster::raster(x = "../../geodata/dem/dem_alos_crop.img")
```





```
## create distance maps
D <- spatial_distance(stations = stations[,3:4],
                      dem = dem)
```

```
## save distance maps
save(D, file = "distance_data.rda")
```

The second step in locating a seismic source is to pass the envelopes of the seismic signals of an event to the function `spatial_distance`, which further requires the distance data calculated before as well as the apparent seismic wave ve-

locity and the sampling frequency (e.g., as denoted in the meta data part of the eseis objects). The apparent wave velocity is the average velocity with which the seismic signals propagate along the surface. Finding a meaningful estimate of this value is crucial and can significantly influence the resulting seismic source location estimate (e.g. Burtin et al., 2013; Dietze et al., 2017a). The function output is a raster object with the average distance-corrected cross correlation coefficient for each pixel.

```
## calculate envelopes of the signals
```
```
e <- signal_envelope(data = s)
```

```
## locate event source
xy <- spatial_migrate(data = e,
                      d_stations = D$stations,
```
```
                      d_map = D$maps,
                      v = 900)
```

```
## clip location estimates to those above 0.99 quantile
xy_clip <- spatial_clip(data = xy,
```
```
                        quantile = 0.99)
```

Usually, one wants to reduce the location estimates to those higher than a given threshold quantile, which can be done using the function `spatial_clip`. The result can then be plotted, for example onto a hill shade (`hs`) map of the terrain.

```
## plot hillshade and location estimate on top
plot(hs,
```
```
     col = grey.colors(200),
     legend = FALSE)
```

```
plot(xy_clip,
```





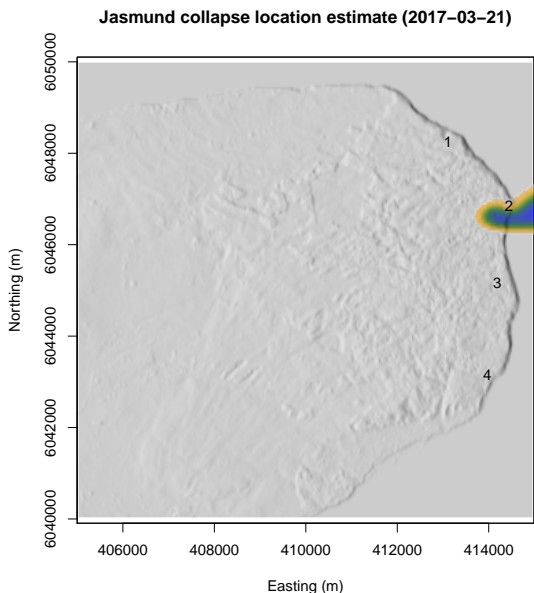

**Figure 7.** Seismic location estimate of the cliff collapse event depicted in fig. 5 as semi-transparent hill shade map overlay with seismic station locations indicated by numbers.

```
        add = TRUE,
        col = adjustcolor(col = rainbow(100),
                          alpha.f = 0.7),
        legend = FALSE)

## add station locations to map
points(x = stations$x,
       y = stations$y,
       pch = as.character(1:4))
```

10     This exemplary work flow showed the basic demands for environmental seismology research questions and how they can be approached with the 'eseis' package. Most of the described functions have further arguments to customise the data processing and manipulate numeric and graphical output. For further details about the functions see the package documentation. For general R-related details about handling objects, passing arguments, and optimising plot output see the generous amount of available tutorials and text books (eg. Adler, 2012; Albert and Rizzo, 2012).





## 4.2  Modelling fluvial dynamics

Rivers cause seismic signals due to transported particles impacting the bed, waves at the water surface, cavitation and turbulent flow (Gimbert et al., 2014). For the latter, dominant source, Gimbert et al. (2014) have developed a physical model. This model has been translated to R by Sophie Lagarde and became a part of the 'eseis' package. It calculates a seismic power spectral density estimate based on parameters describing the fluid, topography and bedrock/sediment properties as well as a series of seismic boundary conditions, in total a set of 23 function arguments.

Here I show how the model can be applied and illustrate the flexibility of the package when combining 'eseis' functionalities with generic R capabilities. The model is used to predict the water level of the Wernersbach, a small river in the Tharandter Wald, near Dresden, Germany, through a period of about three days during which a small flood occurred. The Wernersbach is a typical upland river with a catchment size of 4.6 km$^2$ and average slopes of 3 %. The bed is composed of sand and pebbles, the adjacent terrace contains loamy to sandy Quaternary deposits. The stage has been monitored independently by optical sensing (Eltner et al., 2017). Hourly precipitation data from a meteorological station in Wilsdruff, about 6 km to the North, was exploited using the package 'rdwd' (Boessenkool, 2017) to check for contamination of the seismic signal by raindrop impacts. A Nanometrics Trillium Compact 120s broadband seismometer and Nanometrics Centaur data logger, recording at 400 Hz with 40 V input range provide the empiric seismic data set.

The empiric data were imported as hourly signal traces of the vertical component with a both sided buffer of 300 s, de-convolved, detrended, filtered between 5 and 190 Hz and clipped to the full hour to account for edge effects. The spectra were calculated with `method = "autoregressive"` to receive smoothed estimates, and the spectral power values for frequencies between 10 and 100 Hz were stored for further analysis. This frequency window was decisively used to ovoid low frequency contamination of the record by other than fluvial dynamics.

The turbulence model is a deterministic one. However, when run through R it is easy to include uncertainty in model parameters by a Markov Chain Monte Carlo approach. For this, all relevant model parameters were randomly resampled $10^4$ times according to the boundaries given in table 2. This approach was applied to a series of potential water levels (from 0.01 to 2.00 m in steps of 0.05 m), yielding a set of 60000 potential spectra that served as a lookup table for the empiric data (figure 8 a). The hourly empiric spectra were compared with the Monte Carlo-based potential model spectra to extract only model solutions with average differences below the 0.05 quantile, i.e. the $> 95$ % best fits to the data. From these solutions the means and standard deviations of the respective water levels were computed and interpreted as possible river stages (figure 8 b).

The respective R code is provided in the supplementary materials with less illustrative additional text compared to the firts example, to show a minimum amount of information needed to document code. However, an inspection of the code reveals how most of the criteria defined in section 2 can be met. The 'magrittr'-based piping operator (`%>%`) is used to efficiently pass the `eseis` object from one preparation function to another. Whenever the functions of the 'eseis' package are not sufficient to realise the envisioned analysis step, generic R is used. In this case, the spectra are clipped to the frequency range of interest (10–100 Hz) and stored as data frame for further use. Likewise, the Monte Carlo-based model spectra are isolated from the eseis objects and converted to a numeric matrix for efficient further calculations.




**Table 2.** Input parameters and value ranges for the turbulence model of Gimbert et al. (2014) as implemented in the 'eseis' package.

| Parameter | function argument name | range | unit |
|---|---|---|---|
| Average sediment size | d_s | 0.001–0.01 | m |
| Sediment size log standard deviation | s_s | 1.0–1.5 | log(m) |
| Sediment specific density | r_s | 2600–2700 | g cm$^{-3}$ |
| River width | w_w | 1.0–1.5 | m |
| River height | w_h | 0.01–2.00 | m |
| River gradient | w_a | 0.03–0.05 | rad |
| Frequency range to model | f | 10–100 | Hz |
| Station distance to source | r_0 | 1.5 | m |
| reference frequency | f_0 | 1 | Hz |
| Quality factor at f_0 | q_0 | 5–10 | dimensionless |
| Group velocity | f_0 | 800–1000 | m s$^{-1}$ |
| Quality factor increase exponent | p_0 | 0.40–0.60 | dimensionless |
| Displacement amplitude factors | n_0 | 0.5–0.7, 0.7–0.9 | dimensionless |

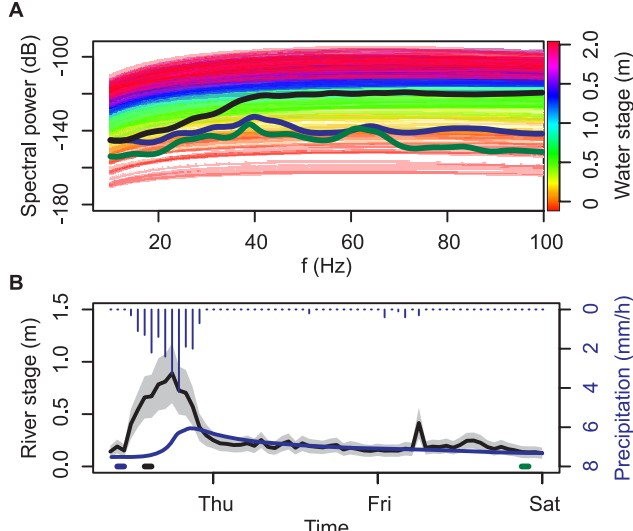

**Figure 8.** Results of the turbulence modelling exercise. a: Monte Carlo-based estimates of spectral power for different water stages at the Wernersbach site together with representative empiric spectra (see panel b for timing of the representative spectra). b: Seismically estimated water levels including parameter-inherent uncertainty ranges (grey polygon and line) together with independent structure-from-motion-based water levels and precipitation data from a meteorological station about 6 km away. Note contamination of the seismic signal of fluvial turbulence by precipitation on Thursday and peak in seismic power due to station maintenance on Friday morning.





When inspecting the model output together with the other data (figure 8) it is evident that the turbulence model by Gimbert et al. (2014) provides a fair estimate of the independently constrained water level of the instrumented river even though the deployment conditions (distance to source, dense tree cover that effectively transmits wind energy into the ground, Dietze et al. (2015)) and river boundary conditions (river size and total water runoff, river course far from straight, relative water

level change during the flood) are far from favourable from a seismic and modelling perspective. For most of the modelled time period the independent water level estimate overlaps with the uncertainty range of the seismic model, except for the late hours of Wednesday, 26 July 2017, when the rain storm that triggered the small flood contaminates the seismic signal by drop impacts (see Turowski et al. (2016) for an example of the seismic signature of rain) and the maintenance visit of a close-by stream gauge early Friday.

**5    Avenues of package development**

The 'eseis' package as described in this article is merely a snapshot of the current state of development and projecting the recent activity due to adding new functions, implementing optimised routines, and feature enhancements and bug fixes due to feedback from package users, it is hard to define a level, which may be interpreted as development plateau. However, there is a series of obvious features that should be included to expand the applicability of the package. Including these features

would be significantly facilitated by expanding the number of scientists that contribute functions and interact with the package maintainer.

**5.1    Anticipated future functionalities**

The current functionality of the package is devoted to analysis of the signals actively emitted by Earth surface processes. An entirely different field of application is monitoring the changes in the properties of the material through which the seismic

waves travel. Bedrock and sediment cover are not static parts of the system, they change by, for example different ground water levels, freeze-thaw transitions, propagation and closure of cracks and reversible/irreversible compaction. Coda wave interferometry or seismic noise cross-correlation (e.g., Sens-Schönfelder and Larose, 2010; Larose et al., 2015) is a powerful method to tackle this field. Thus, integrating the basic calculations to the 'eseis' package is a relevant future task.

Location of environmental seismic sources can be performed by several approaches, of which the signal migration technique

is just one. Other aproaches rely on cross-correlating signals of station pairs and identifying the wave train azimuts that best explain the correlation offsets, i.e., beam forming (e.g., Lacroix and Helmstetter, 2011). For strong signals with clearly spearated wave types, polarisation analysis allows for finding the azimut from which a signal approaches a seismic station (Jurkevics, 1988; Vilajosana et al., 2008). Apart from signal correlation approaches also the amplitude of signals as recorded across a seismic network can be used to find a location that best explains the exponentially decreasing amplitudes with distance from

source to station (Aki and Ferrazzini, 2000; Burtin et al., 2016). Most of these location approaches were formulated in a script language and can thus in principle be translated to R to become a valuable part, increasing the applicability and flexibility of the package.




Physically based or empirical scaling models for other than fluid turbulence become more and more available (e.g., Tsai et al., 2012; Farin et al., 2015) and will allow tackling innovative research questions, especially when combined with R-provided methods to account for the inherent uncertainty of the model parameters, way beyond the simple example from section 4.2. Integrating such models to the 'eseis' package is a step that has already been started and that will continue.

A key functionality for a wider acceptance of the 'eseis' package will be its role in importing and exporting a wider range of data formats. Although with the sac and mseed format the two most commonly used formats are supported – at least for import – there are other file formats in use that should be included. One potential way of reaching this goal without dublicating already existing work would be to use the Obspy capacities for reading and writing data by integrating the respective modules into the compilation part while installing the package or by simply wrapping the Python code in R functions, which would
make a local Obspy installation a requirement.

## 5.2 Getting involved

The 'eseis' package will and can only evolve along emerging needs of researchers. Thus, from a user perspective, the most important avenue to contribute is to report bugs, provide feedback about missing features and raise ideas on how to improve functionalities and applications.

In R, the transition from user to developer is soft and does not include a steep learning curve. This is one of the reasons why R become so widely accepted and used (Tippmann, 2014). Thus, contributing ideas in the form of R code, may it be models or data analysis functions, is highly welcome and would further improve the package development. R has a refined approach to assign roles to package development collaboration teams, which guarantees acknowledgement and transparency of the persons contributing to individual items of a package (Team, 2016).

## 6   Conclusions

The 'eseis' package provides functions to import, prepare, manipulate, analyse, visualise and export seismic data. Thus, it contains all functionalities to engage with seismic data in general and environmental seismology topics in specific. It is not intended to replace existing open software such as Obspy that is in several fields more powerful and efficient. Rather, it is tailored to bridge different environmental scientific fields at a low level of complexity and utilising an easy to learn and yet
broadly accepted scripting language. It contributes to this bridging effect by guaranteeing support of and passing data to other, differently specialised R packages.

The package will live from interaction between users and developers, and users eventually becoming developers – a common effect among the R community. Thus, the package maintainer deliberately welcomes feedback, suggestions to improve and broaden the functionality, and to fix bugs and drawbacks.
One key but hitherto not fully agreed precondition to transparent and reproducible science demands also citing software and documenting the actual software environment. Citing for example R packages appropriately (a job that can be easily done with the R function `citation("package_name", auto = TRUE)`) does not only acknowledge the effort scientists



put into providing such tools for a broad audience. More importantly, it assures the reproducibility of scientific results. This citing culture includes at best information about the computer platform and key system parameters, the version of the software and packages used, and the analysis chain (including parameter values). The 'eseis' package provides a straightforward support for this with the `write_report` function. Combined with a version control software, such as Git, the full time line of data

5  processing and analysis script evolution can be documented easily.

## 7  Data and code availability

During the publication process, the R package 'eseis' will be made available both on the Comprehensive R Archive Network (CRAN) and on Github.

*Author contributions.*  Michael Dietze wrote the R package 'eseis', evaluated the worked examples and wrote the manuscript text.

10  *Acknowledgements.*  The author of this article wishes to thank Anette Eltner for deliberately providing the optical water level data, Stefanie Puffpaff, Ingolf Stodian and the Jasmun/Rügen National Park team for the seamless collaboration and data assistance, Arnaud Burtin for essential discussion at early stages of the project and insight to his scripts, Sophie Lagarde for her efforts in writing up the turbulence model, Anne Schöpa and Kristen Cook for helpful feedback on the package performance and discovering painful bugs, Trond Ryberg and Christoph Lendl for the patience with my demands, Karl-Heinz Jäckel for enlightening discussions about sensors and data handling, Omnirecs for

15  providing tailored instruments, and the R community for the marvellous range of just perfect packages.




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
