# Peer review of "The R package 'eseis' – a software toolbox for environmental seismology"

_Earth Surface Dynamics, 2017_

## Referee Comment (RC1) · Anonymous Referee #1 · 20 Feb 2018

The article describes an innovative toolbox package to process seismic data for environmental, trans disciplinary purposes. It is too early to determine whether the package will be of major help for the community, but this article will be certainly important to initiate developments and attract interest. The article is well written and well documented, and significantly informative about the use of the package. After minor corrections/complements, I suggest to publish the manuscript in Earth Surf Dyn

Minor comments - do you include readings of other formats like SE2 or SEGY ? - do you provide a user manual ? - when processing data from different stations, how do you handle heterogeneous lacks of samples and sample redatings ?

In general, the package does not currently process ambient noise xcorr, which is a major field of environmental seismology. Maybe you should change the title accordingly,

like Âń The R package 'eseis' – a comprehensive software toolbox for source analysis and active environmental process'Âż and restrain the current paper to studying active source process?

P3: you pretend that R will be stable on the long term. How can you make sure of this ? many other languages pretended the same years ago, some are still maintained, others have retired. . .

P4 , pat 3.1 : how will you handle file format for correlations data (from ambient noise for instance)?

P6: tools proposed for Centaurs and Cubes stations. Why those ones and not others ? If this is just a choice due to hardware availability at the authors' home, please mention it.

P7 L14: higher order of 2n = ? please explain literally that it's the total number of samples, with n a positive integer. . .

P9 fig 4: verb tenses ?

P10 L5: make sure the example dataset is an actual rockfall and not a base-jump crash !

P16 L25-30: please specify clearly form the beginning that the source location is obtained for surface waves (and not P-wave migration)

P21 fig 8: curves (b) are not labeled/commented; the figure is overall impossible to understand. What is what ?

P23 5.2: please recall an active url. . .

Please change ref SensSchoenfelder and Larose Earthquake Science 2010 by C. Sens-Schönfelder and E. Larose : Temporal changes in the lunar soil from correlation of diffuse vibrations, Phys. Rev E 78, 045601 (2008).

---

## Referee Comment (RC2) · Anonymous Referee #2 · 25 Apr 2018

This paper presents a new data analysis software package called " e-seis " dedicated to the processing of seismic data and intended to be mostly used to analyze seismic signals associated with various surface processes. This library written in R aims at facilitating the routine tasks for new or advanced seismologists, including reading seismic data in various popular formats, some common preprocessing steps (filtering, instrumental response removal, ...), standard plotting (spectrogram, ...) or basic analysis (phase picking based on STA/LTA, ...). Some more advances tools are also available to analyses the non-impulsive seismic signals often associated with surfaces processes.

Overall it is a well written paper which includes various didactic portions of code and working examples. Although not being personally familiar with R, the "eseis" package

seems to be well organize and quite straightforward to use, at least based on the few examples given by the author.

My main concern is mainly about the introduction part of the paper and the motivations for the "eseis" package : - The author has to better explain his motivation for developing this new R package even through there exists several other seismic data (pre)processing and analysis open source solutions, among them (Obspy, Seisan, SeismicHandler, . . .) several are intensively used by a broad community. Basically I did not find in the introduction clear answers to some simple questions like : 1) If I a am new to seismic data processing (and/or if I analyze seismic data related to surface processes), why should I use "eseis" rather than another solution ? 2) what is specific to "eseis" compared to other packages (pros/cons)? 3) if I am a "R-lover", why should I use "eseis" rather than the RSEIS package (https://cran.rproject.org/web/packages/RSEIS/index.html)?

- The author argues in the introduction that it is "essential to find a common language" in environmental seismology. Why such a statement? Seismic data used for environmental seismology are not different than other (passive) seismic data and are (pre)processed in similar ways (this is actually what mostly does "eseis"). "Classic seismologists" (not focusing on surface and subsurface processes) also work on non impulsive sources ("eg. tremors"), ambient noise . . . For example, the ambient noise interferometry approach mentioned by the author (p2, l.16-18) is applied in a wide range of seismological studies, including environmental seismology but not only ! And the MSnoise package (http://www.msnoise.org/) already does the job quite well!

- For me, the introduction is too much an apologia for the R language. Other languages, and especially Python, are almost not cited although they are used by a wide and growing range of seismologists (and scientists) having the same motivations as the author.

My advice is to be less ambitious in the introduction avoiding too general (or oriented)

statements and to present the "eseis" package as a promising R solution for easy data processing with some specific modules (that users will not be able to find in other solutions; like the model_turbulence module) dedicated to the analyses of environmental surface sources that produce seismic signals. In the following sections, it might also be good to better separate the aspects related to "standard" processing of seismic data (including preprocessing, temporal/spectral plotting, sta/lta, . . .) from modules purely dedicated to the analysis of surface processes.

Some other more minor or technical comments:

- Part 2 (p2 l21 to p3 l21) could be condensed. Although I agree with most of the author's statements related to data/code sharing policy and general principles in coding, these problematics are for most beyond the scope of this paper.

- Header of the "eseis" objects : How "eseis" is handling the fact that SAC files and miniseed files do not have the same information in their headers (SAC being more event oriented whereas miniseed is more dedicated to continuous streams) ? Is it possible to add information in the headers (such as events information)?

- Low level programming languages (p5, l11-15) : Note that there are a lot of other analysis techniques, not yet developed in "eseis", which would benefit from the use of low level languages (for example for continuous scanning of waveform parameters)

- Data structure (p6 l20) : The Year/Julian day file structure is not so common. Lot of seismologists use a Seiscomp "Standard Data Structure (SDS)" (Year/Net/Sta/Chan) with day long files.

- Metadata : It seems that "eseis" does not have the ability to read/write standard FDSN metadata formats (Seed dataless, StationXML). They are used by a wide variety of seismologists and they include information that may be crucial for some processing. If "eseis" does not accept such type of metadata formats, the author should mention the implications and potential limitations of their dedicated way of handling metadata.

- Deconvolution (p7 l4-10) : following the previous comment, it seems that the digitizers have only a "gain" parameter. Not taking into account stages such as anti-aliasing filter coefficients may lead to some misinterpretation of the signal in time or frequency domain.

- Metadata / channel naming: Legend 1 indicate all the "relevant metadata" but I don't see information like the "location code", "channel name", the orientation of the sensor, etc... For example how works the signal_rotate module (or others) if the orientation of the components are not provided in the input metadata file ?

- Output format (p9 l4) : Indicate what is the main reason for choosing SAC as the output format.

---

## Author Comment (AC1) · 1 May 2018

[11pt, a4paper]article pdfpages hyperref listings

**Response to referee comments**

[The R package 'eseis' – a software toolbox for environmental seismology]
May 1, 2018

I would like to thank the referee for the encouraging and helpful comments, all of them

obviously devoted to improve the quality and impact of the manuscript.
* * *
*Referee 1.1: do you include readings of other formats like SE2 or SEGY?*

**Reply**: Currently, the only supported formats are SAC and miniseed. I explicitly mention this now earlier (p. 6, l. 7-8) than just in the dedicated section 5.1 and indicate that if required by users, import of additional data formats can be coherently implemented.
* * *
*Referee 1.2: do you provide a user manual?*

**Reply**: Yes, a user manual (reference manual) is automatically generated when building the R package. This is now mentioned in chapter 3 (p. 6, l. 1-2).
* * *
*Referee 1.3: when processing data from different stations, how do you handle heterogeneous lacks of samples and sample redatings?*

**Reply**: Like in most other approaches, such challenges must be handled individually. However, the object philosophy of the package always allows direct access to the data set. Thus, one can "easily" handle missing data, NA values or develop methods fo redating in R. I mention this issue now section 3.1 (p. 5, l. 23-25).
* * *
*Referee 1.4: In general, the package does not currently process ambient noise xcorr, which is a major field of environmental seismology. Maybe you should change the title*

*accordingly, like "The R package "eseis" – a comprehensive software toolbox for source analysis and active environmental process" and restrain the current paper to studying active source process?*

**Reply**: In fact, ambient noise cross correlation is currently being implemented to the package as indicated in section 5.1. Thus, in one of the upcoming versions of the package (in a few months probably), this feature will be included. However, I understand the concern and removed the word "comprehensive" from the title, as it is clear that the package in its current form does not include all major avenues of (environmental) seismology.
* * *
*Referee 1.5: you pretend that R will be stable on the long term. How can you make sure of this? many other languages pretended the same years ago, some are still maintained, others have retired...*

**Reply**: Ha, very good point indeed. One may only use the past to predict something about the future but nothing is certain for sure. I added this topic and refer to the code of conduct of the R foundation (the consortium that has formed some years ago to cover exactly this topic) in section 2 (p. 3, l. 8-11).
* * *
*Referee 1.6: how will you handle file format for correlations data (from ambient noise for instance)?*

**Reply**: Just similar to point 1.3, the user can do with the data (in this case the signal vector part) whatever she or he wants to do. It is, at least within R, even possible to store e.g., complex values in the signal vector. So no problems in that respect. From

my perspective a good format to save correlation data would be as compressed R data (*.rda) objects. Anyhow, since ambient noise cross correlation analysis is not yet part of the package I would prefer not to explicitly mention this. I did however insert a short general sentence in section 3.1 (p.5, l. 25) highlighting that any derivatives of the input data can be handled and stored as R objects.
* * *
***Referee 1.7****: tools proposed for Centaurs and Cubes stations. Why those ones and not others? If this is just a choice due to hardware availability at the authors' home, please mention it.*

**Reply**: I do now provide the reason for the support of these two loggers (predominantly because these are the ones I have used in my projects so far and therefore can rely on ample benchmark and test data sets) but also refer to the option of additional logger/data structure incorporations in future package versions if users of the R package express their need for that. I am happy to develop such extra functionalities, at best in cooperation with the future users.
* * *
***Referee 1.8****: higher order of 2n = ? please explain literally that it's the total number of samples, with n a positive integer...*

**Reply**: Done as suggested.
* * *
***Referee 1.9****: fig. 4 verb tenses?*

**Reply**: Corrected.

**Referee 1.10**: *make sure the example dataset is an actual rockfall and not a base-jump crash!*

**Reply**: Indeed, I checked for that cause of a seismic source (http://base-jumping.eu/base-jumping-fatality-list/) but could not link the two possible cases to any of the locations in the original Lauterbrunnen rockfall study.

**Referee 1.11**: *please specify clearly form the beginning that the source location is obtained for surface waves (and not P-wave migration)*

**Reply**: Done as suggested, mentioned now on p. 17, l. 20.

**Referee 1.12**: *curves (b) are not labeled/commented; the figure is overall impossible to understand. What is what?*

**Reply**: Figure has been reworked. A legend is now present in the figure and the two PSD are linked to the time series plot.

**Referee 1.13**: *5.2 please recall an active url*

**Reply**: Done as suggested.

**Referee 1.14**: *change ref SensSchoenfelder and Larose Earthquake Science 2010 by C. Sens-Schönfelder and E. Larose: Temporal changes in the lunar soil from correlation of diffuse vibrations, Phys. Rev E 78, 045601 (2008).*

**Reply**: Done as suggested.

---

## Author Comment (AC2) · 1 May 2018

[11pt, a4paper]article pdfpages hyperref listings

**Response to referee comments**

[The R package 'eseis' – a software toolbox for environmental seismology]
May 1, 2018

I would like to thank the referee for the encouraging and helpful comments, all of them

obviously devoted to improve the quality and impact of the manuscript.
* * *
*Referee 2.1: My main concern is mainly about the introduction part of the paper and the motivations for the "eseis" package: - The author has to better explain his motivation for developing this new R package even through there exists several other seismic data (pre)processing and analysis open source solutions, among them (Obspy, Seisan, SeismicHandler,...) several are intensively used by a broad community. Basically I did not find in the introduction clear answers to some simple questions like: 1) If I a am new to seismic data processing (and/or if I analyze seismic data related to surface processes), why should I use "eseis" rather than another solution? 2) what is specific to "eseis" compared to other packages (pros/cons)? 3) if I am a "R-lover", why should I use "eseis" rather than the RSEIS package (https://cran.rproject.org/web/packages/RSEIS/index.html)?*

**Reply**: I agree and see some of the arguments fully justified. I clarified the introduction, especially the second paragraph, in several sections to address where: i) R differs from other software commonly used in seismology, ii) that the approach is not "from seismology to environmental disciplines" but rather the other way around: "opening the door to utilising seismic data for a diverse range of disciplines that are very much used to working with R" and iii) one would benefit from working consistently in one software environment rather than processing and routing data from one isolated software with its specific syntax or GUI to another. Hence, the pros – or justifications – are elaborated on in the introduction but also section 2 and 3.1. The drawbacks are now pointed out in section 3.2, during the descriptions of each of the data processing steps, as demanded in the referee's points 2.7, 2.9, 2.10, 2.11 and 2.12.

With respect to the R package 'RSEIS', my package differs fundamentally in the way

data is handled as eseis objects, how the processing chain is organised and implicitly documented, and how the package is managed. I did not find a conflict-free way in the manuscript to point at the comparably "less well organised" collection of functions (several appear to be doubled but with different names and partly different output structure), not to speak of the documentation of and examples for each function. The best recommendation would be, please try to work with alternatives to 'eseis' in R and report your experiences, but this is, again, not a polite and helpful phrase in a manuscript I think. Actually I benefited a lot from working myself through the source code of many 'RSEIS' functions but it is an experience I would like not everyone to share. I added a more elaborated sentence just before opening section 3.1 (p.4, l. 17-19) to clearly point out in which dimensions 'eseis' differs from other R packages devoted to seismology.
* * *
*Referee 2.2: The author argues in the introduction that it is "essential to find a common language" in environmental seismology. Why such a statement? Seismic data used for environmental seismology are not different than other (passive) seismic data and are (pre)processed in similar ways (this is actually what mostly does "eseis"). "Classic seismologists" (not focusing on surface and subsurface processes) also work on non impulsive sources ("eg. tremors"), ambient noise... For example, the ambient noise interferometry approach mentioned by the author (p2, l.16-18) is applied in a wide range of seismological studies, including environmental seismology but not only! And the MSnoise package (http://www.msnoise.org/) already does the job quite well!*

**Reply**: I completely agree and think it is a misunderstanding. I did not mean to imply it is essential to find a common language in environmental seismology. Rather, the common language needs to be arranged among the scientific disciplines (geomorphology, hydrology, meteorology, glaciology, and so on) with respect to the seismic approach. And the term language refers to both, jargon and programming language. I explicitly mention the scope of the statement now in the text (p. 2, l. 4).
*Referee 2.3: For me, the introduction is too much an apologia for the R language. Other languages, and especially Python, are almost not cited although they are used by a wide and growing range of seismologists (and scientists) having the same motivations as the author.*

**Reply**: I now explicitly mention the role of Python in seismology (p. 2, l. 1) and would as well refer to point 2.1., where I clarify that the idea of R and the 'eseis' package is about "opening the door to utilising seismic data for a diverse range of disciplines that are very much used to working with R" instead of "yet another software solution for seismologists interested in study Earth surface dynamics". In order to contribute to this topic in addition to my replies to point 2.1, I gave more emphasis on the two references that provide examples of available tools for seismology (p. 1, l. 24).

*Referee 2.4: My advice is to be less ambitious in the introduction avoiding too general (or oriented) statements and to present the "eseis" package as a promising R solution for easy data processing with some specific modules (that users will not be able to find in other solutions; like the model_turbulence module) dedicated to the analyses of environmental surface sources that produce seismic signals.*

**Reply**: I assume my replies to points 2.1-2.3 cover this topic.

*Referee 2.5: In the following sections, it might also be good to better separate the aspects related to "standard" processing of seismic data (including preprocessing, temporal/spectral plotting, sta/lta,...) from modules purely dedicated to the analysis of surface processes.*

**Reply**: I inserted a paragraph to address this topic (p. 11, l. 18-20).
* * *
*Referee 2.6*: Part 2 (p2 l21 to p3 l21) could be condensed. Although I agree with most of the author's statements related to data/code sharing policy and general principles in coding, these problematics are for most beyond the scope of this paper.

**Reply**: I am not sure about the statement that the problematics are beyond the scope of my paper. I would argue that they in fact are the motivation and justification of the R package and working with free and open software in general. Section 2 is arranged in a way that each paragraph addresses a different dimension of open and reproducible science and they feed into details explained in section 3.1. I initially had the material from section 2 organised in the introduction but realised that it would have caused a mismatch in focus (and amount of words for each of the sub topics of the introduction) as it makes up a section on its own. I would prefer to keep the section in its current form.
* * *
*Referee 2.7*: Header of the "eseis" objects : How "eseis" is handling the fact that SAC files and miniseed files do not have the same information in their headers (SAC being more event oriented whereas miniseed is more dedicated to continuous streams) ? Is it possible to add information in the headers (such as events information)?

**Reply**: Yes, it is always possible to modify, add or remove elements of R objects, including the header and meta elements. Thereby, the header element is the original material imported from the seismic data file, and the meta element is another list that is generated for R-internal purpose, to consistently work with the 'eseis' object. However, since the package currently allows only to write SAC files, and mseed and SAC

are not fully comparable there might be some trade off. I mention now explicitly this point in section 3.2.1 (p. 6, l. 15-17).
* * *
***Referee 2.8****: Low level programming languages (p5, l11-15) : Note that there are a lot of other analysis techniques, not yet developed in "eseis", which would benefit from the use of low level languages (for example for continuous scanning of waveform parameters)*

**Reply**: The misleading sentence has been removed. I was not about to say that low level code should not be used, for example in future package functionalities. Of course, whenever there is good reason for implementing code of another language this shall and can be done in R.
* * *
***Referee 2.9****: Data structure (p6 l20) : The Year/Julian day file structure is not so common. Lot of seismologists use a Seiscomp "Standard Data Structure (SDS)" (Year/Net/Sta/Chan) with day long files.*

**Reply**: Very good point. I added this to the list of feature requests for upcoming versions of the package.
* * *
***Referee 2.10****: Metadata : It seems that "eseis" does not have the ability to read/write standard FDSN metadata formats (Seed dataless, StationXML). They are used by a wide variety of seismologists and they include information that may be crucial for some processing. If "eseis" does not accept such type of metadata formats, the author should mention the implications and potential limitations of their dedicated way of handling*
*metadata.*

**Reply**: Indeed, these meta data formats are not yet part of the package. I now explicitly mention this at the end of section 3.2.1 (p. 7, l. 22 – p. 8, l. 2) along with the consequences. Again, for a future version of the package it is envisioned to provide such support as it is obviously a vital goal to increase the acceptance of the package in different communities, as discussed in the dedicated section 5.1.
* * *
*Referee 2.11: Deconvolution (p7 l4-10) : following the previous comment, it seems that the digitizers have only a "gain" parameter. Not taking into account stages such as anti-aliasing filter coefficients may lead to some misinterpretation of the signal in time or frequency domain.*

**Reply**: The logger list contains information as noted below. So with respect to the deconvolution the digitizers are characterised by their AD value, while the gain parameter is set as argument of the deconvolution function. But the referee is right, no filter coefficients etc. are accounted for, at the moment. Thus, I mention this point in the text (p. 8, l. 10-11) and would refer to point 2.10 that with upcoming package versions and dataless seed and StationXML support this shall be resolved.

```
eseis::list_logger()$Cube3ext
$ID
[1] "Cube3ext"
$name
[1] "Cube 3ext"
$manufacturer
[1] "Omnirecs"
```

```
$type
[1] "n.n."
$n_components
[1] 3
$comment
[1] ""
$AD
[1] 2.4414e-07
```
* * *
*Referee 2.12*: *Metadata / channel naming: Legend 1 indicate all the "relevant meta-data" but I don't see information like the "location code", "channel name", the orientation of the sensor, etc. For example how works the signal_rotate module (or others) if the orientation of the components are not provided in the input metadata file?*

**Reply**: The figure shows the actual output of the function `write_report()` when applied to an imported seismic file. What is shown is the meta data (not the header data). And in this case there was no network code provided to the Omnirecs Cube logger config file. Likewise, the imported SAC file did not contain any information about location or sensor and logger type in its header part. Thus, the import assigned NA values for these parameters. The channel name is represented by the element "component" (here "p0"). I added a clarifying sentence pointing at the missing information and how it can be documented (and any changes traced) by these reports (p. 9, l. 11-13).

With respect to the second example question, the function `signal_rotate()` has predefined argument values for the channel order that can be changed manually if needed. All these information are given in the function documentation manual or online help.

---

## Author Response (AR2)

**Response to referee comments**

[The R package 'eseis' – a software toolbox for environmental seismology]
June 12, 2018

I would like to thank the referee for the second round of comments. I agree with all of them and have implemented the requested changes as suggested.
* * *
**Referee 2.1**: *Overall this new version of the paper has been greatly improved. I thank the author for the clear answers to my first comments and for the substantial modifications in the manuscript. All the technical points I raises have been answered and well clarified in the text, especially regarding the main limitations and potential evolutions of the "eseis" package. My only remaining concern is still related to part 1 and 2. I am still uncomfortable with the way the R language and the "eseis" package are introduced. To my opinion, the author is emphasizing too much the need for a common language and seems to consider that R is the only appropriate common language ("it is essential to find a common language", "This common language should be", "R can serve as the language", ...). Why not considering the R language as one of the suitable language (as is Python) in the geosciences domain? Why such a long emphasis about a common language? One can also consider that the emphasis should rather be on common data format, reproductivity, code versioning, ... but not the language for the scripting of the codes.*

**Reply**: I changed the introduction in several sections. Specifically, the following sentences were changed:

"Furthermore, since environmental seismology integrates several neighbouring and more distant scientific fields, to which the seismological results are passed as input data, it is essential to find a common language among these scientific fields, a language that is not necessarily driven by a seismic background. This common language constraint applies to both, the scientific jargon and the data analysis software language." has been removed.

"While the language Python is widely used in seismological research, and partly also in disciplines like remote sensing, terrain analysis and climatology, there is another software language tailored to data science that also fulfils all these qualifiers: the free statistic software R (RCoreTeam, 2015)." has been replaced by "The free statistic software R is one example of a software that fulfills these qualifiers. "

"Thus, R can serve as the language for integrating disciplines and providing these disciplines with methods to utilise seismic data, hence allowing research with one software environment rather than passing data and intermediate results from one isolated software to the next." has been removed.

"Ideally, this common language" has been replaced by "Ideally, a software used for processing data with a broad context of application"
* * *
**Referee 2.2**: *Also, there is a little bit of a contradiction when the author argues in the introduction for a common language across various disciplines (including seismology), namely targeting the R language, and later states (p6 - l3; p24 - l1, p24 - l16, ...) that other analysis tools used in seismology (SAC, Obspy), written in other languages, are more adapted for a wide variety of tasks than "eseis" and should therefore also be used (I fully agree with that). This is rather incoherent with the statement that the community (or "communities") should go toward a common language. Or, does the author want to express the opinion that in the future we should give up with these existing and quite performant tools and convert them into R codes?*

**Reply**: Correctly identified contradiction. I think the changes documented for point 2.1 should have resolved this issue. And also right, I do not imply we all should switch from tools well established in given communities to another software/language.
* * *
**Referee 2.3**: *Therefore, I still suggest being less "ambitious" in the introduction and to focus more on the main objective of this contribution which is to present the "eseis" package. I would advise to present, in a much shorter way, why "eseis" has been written in R (giving some of the pertinent arguments about the advantages of the R language) and put much less emphasis on the need for a common language.*

**Reply**: I think this also relates to the requested changes demanded in point 2.1. Significant shortening and generalisation has been implemented.
* * *
**Referee 2.4**: *The discussion about the potential need for a common language, and the fact that R might be THE one (although I personally consider that Python can also be  but I am perhaps "too much" a seismologist) is of great interest but should be expressed by the author in another dedicated "opinion paper".*

**Reply**: Perhaps the text of the previous versions was too misleading. My intention was not to render all other (open) software unsuitable for environmental seismology. So perhaps an opinion paper might pick up some of these ideas, but should not revolve around "R is the only language". Anyhow, I am grateful for the idea and will think about it.
* * *
**Referee 2.5**: *P2  l9 : "R [...] is yet exponentially growing in terms of users and provided packages ..." Please check that the number of users is indeed growing exponentially. The few curves I found on the web rather show a linear trend during the past about 10 years.*

**Reply**: Indeed, information about such trends is hard to find. My statement was based on a talk during the UseR conference in Brussels last year (https://www.user2017.brussels/schedule), focusing on such statistics. However, I agree that internet searchers for such plots yields limited results. Thus, I rephrased the sentence, replacing "exponentially" by "continuously".